# A Sweet Cherry Glutathione S-Transferase Gene, PavGST1, Plays a Central Role in Fruit Skin Coloration

**DOI:** 10.3390/cells11071170

**Published:** 2022-03-30

**Authors:** Xiliang Qi, Congli Liu, Lulu Song, Yuanxin Dong, Lei Chen, Ming Li

**Affiliations:** Zhengzhou Fruit Research Institute, Chinese Academy of Agricultural Sciences, Zhengzhou 450009, China; qixiliang@caas.cn (X.Q.); liucongli@caas.cn (C.L.); songlulu2010@163.com (L.S.); d18734723812@163.com (Y.D.); sparenoefforts0224@163.com (L.C.)

**Keywords:** sweet cherry, anthocyanin accumulation, glutathione S-transferase, PavGST1, PavMYB, promoter

## Abstract

Sweet cherry, an economically important horticultural crop, has strong antioxidant activity. The fruits contain compounds potentially beneficial to human health—particularly anthocyanins, which are synthesized in cytosol and predominantly accumulated in vacuoles. Although anthocyanin levels differ among dark-red, blush, and yellow sweet cherry cultivars, the regulatory mechanism of anthocyanin transport and accumulation is not well understood in this species. In this study, we identified 53 glutathione S-transferase genes (PavGSTs) from sweet cherry and found that PavGST1 expression was well correlated with anthocyanin accumulation in cultivars with different fruit skin colors. TRV-mediated virus-induced silencing of PavGST1 decreased anthocyanin accumulation in sweet cherry fruits and downregulated the expressions of anthocyanin biosynthetic and regulatory genes. In addition, transient overexpression of PavGST1 promoted anthocyanin accumulation. Furthermore, yeast one-hybrid and dual-luciferase assays revealed that PavMYB10.1 and PavMYB75 directly bind to different MYB binding sites of the PavGST1 promoter (MBS-1 and MBS-3) to activate PavGST1 transcription. According to our results, PavGST1 plays a central role in sweet cherry fruit anthocyanin accumulation. Our findings provide novel insights into the coordinative regulatory mechanisms of PavGST1 and PavMYBs in anthocyanin accumulation in sweet cherry.

## 1. Introduction

Anthocyanins are important natural water-soluble flavonoid pigments widely distributed in leaves, fruits, seeds, and flowers that are responsible for red, purple, black, and blue coloration in these plant organs. Anthocyanins carry out diverse biological and physiological functions that aid pollination and seed dispersal, confer biotic and abiotic stress resistance, and prolong fruit life span in plants [1,2,3]. Given their potent antioxidant properties, such as immunomodulatory, antioxidant, cardio-protective, antithrombotic, and anti-cancer activities [4,5,6,7], many anthocyanins are also human health-promoting food ingredients. 

Anthocyanins are biosynthesized by a series of enzyme-encoding structural genes in the flavonoid pathway that uses phenylalanine as the precursor. These genes include early biosynthetic genes encoding phenylalanine ammonia lyase (PAL), chalcone synthase (CHS), chalcone isomerase (CHI), flavanone 3-hydroxylase (F3H), and flavonoid 3’-hydroxylase (F3’H), as well as late biosynthetic genes for dihydroflavonol 4-reductase (DFR), anthocyanin synthase (ANS), and UDP-glucose:flavonoid 3-glucosyltransferase (UFGT), all having specific contributions to anthocyanin biosynthesis [3,8,9,10]. In addition, synthetic anthocyanins in cytosol are transported to vacuoles for storage [11,12,13]. The involvement of three distinct anthocyanin transporters has been proposed for this process: glutathione S-transferase (GST) transporters, ATP-binding cassette (ABC) transporters, and multidrug and toxic compound extrusion (MATE) family transporters [14,15,16,17,18,19].

In this study, we focused on the GST family. Encoded by a large gene family in plants, GSTs play vital roles in various physiological and developmental processes, including resistance to biotic and abiotic stresses, detoxification of xenobiotics and toxic lipid peroxides, glucosinolate biosynthesis and metabolism, and signaling via catalysis of the conjunction of glutathione (GSH) to various electrophilic compounds [20,21,22]. Numerous GSTs have been reported in plants thus far, including 42 in maize [23], 55 in Arabidopsis [21], 79 in rice [22], 61 in orange [24], 90 in tomato [25], 54 in peach [26], 23 in apple [27], and 97 in kiwifruit [28]. In addition, recent studies have functionally demonstrated that some plant GSTs are essential for anthocyanin and proanthocyanidin vacuolar accumulation. The function of these GSTs in vacuolar anthocyanin transport was first ascertained in the maize bronze-2 (*bz2*, GST-like protein) mutant [29], which was deduced by the visible loss of pigmentation. GSTs with similar functions have been subsequently identified in various plant species; examples include transparent testa 19 (*AtTT19*) in *Arabidopsis* [14], anthocyanin 9 (*PhAN9*) in petunia [30], flavonoid 3 (*fl3*) in carnation [31], and *CMGSTF12* in Brassicaceae [32]. Anthocyanin-related GSTs have recently been identified as essential for fruit or floral pigmentation in fruit crops, including *VviGST1* and *VviGST4* in grape [33,34], *LcGST4* in litchi [35], *FvRAP* (encoding a GST transporter for anthocyanin) in strawberry [36], *AcGST1* in kiwifruit [28], *MdGSTF6* in apple [27], and *PpGST1* in peach [37]. In Rosaceae, however, the molecular mechanisms underlying the influence of the GST gene family on anthocyanin transport and accumulation are still unclear.

Sweet cherry (*Prunus avium* L. (Rosaceae)) is an economically important horticultural crop widely cultivated in temperate regions worldwide [38]. Sweet cherry exhibits a variety of fruit skin colors, including yellow, orange, blush, and dark red. These color differences are due to variation in the content of accumulated anthocyanins, which have strong antioxidant activity and potential benefits for human health [38,39]. Previous studies have demonstrated that the PavMYB10.1-PavbHLH-PavWD40 (MBW) protein complex regulates anthocyanin biosynthesis in sweet cherry and that the R2R3-MYB transcription factor *PavMYB10.1* plays a key role in determining different fruit skin coloration patterns [39]. Nevertheless, the molecular mechanisms underlying anthocyanin vacuolar sequestration and accumulation in sweet cherry fruit skin remain unknown, and anthocyanin-related GSTs, which play a vital role in anthocyanin vacuolar accumulation, have not yet been clearly identified in sweet cherry.

In our current study, 53 GST genes were identified from the genome sequence of sweet cherry [40]. Among these genes, *PavGST1* was very strongly expressed and found to be highly correlated with anthocyanin accumulation in sweet cherry cultivars with different fruit skin colors. The role of PavGST1 in the regulation of vacuolar accumulation of anthocyanins in sweet cherry was subsequently characterized using overexpression and silencing approaches. In addition, the silencing or overexpression of *PavGST1* was revealed to alter fruit skin coloration in sweet cherry, thereby indicating the important function of *PavGST1* in anthocyanin accumulation. Further experiments demonstrated that PavMYB10.1 and PavMYB75 can directly bind to the PavGST1 promoter to activate the expression of PavGST1. These findings revealing that PavGST1 encodes the pivotal anthocyanin transporter in sweet cherry provide new insights into the mechanisms underlying anthocyanin accumulation in this important fruit crop species.

## 2. Materials and Methods

### 2.1. Plant Materials

Sweet cherry cultivars Chunlu, Hongshouqiu, and 13–33 were grown in the resource garden of the National Fruit Tree Germplasm Repository, Zhengzhou Fruit Research Institute, Chinese Academy of Agricultural Sciences (Zhengzhou, China). The mature fruits of the three cultivars were harvested at 55 days after full bloom. *Nicotiana tabacum* used in this study was maintained in our laboratory and was grown at 22–25 °C in a greenhouse with a 16 h light/8 h dark cycle and 60–75% relative humidity.

### 2.2. Anthocyanin, Proanthocyanidin, Flavonol, and Total Flavonoid Extractions and Measurements

Anthocyanins were extracted from sweet cherry fruits and quantified according to the method of Shen et al., 2017 [41]. The peels were extracted with 1% pre-cooled hydrochloric acid-ethanol solution on ice, followed by centrifugation at 10,000× *g* for 30 min at 4 °C. Total anthocyanin content, calculated as cyanidin-3-glucoside, was measured by a pH differential method. Proanthocyanidins were extracted from sweet cherry fruits and measured using a previously reported phloroglucinolysis protocol [42,43]. The powder sample (0.5 g) from sweet cherry fruit peel was extracted with 10 mL of anhydrous ethanol. After centrifugation, the precipitate was dissolved in 3 mL vanillin (4% *w*/*v*) and 1.5 mL concentrated hydrochloric acid. Then, the absorbance of the reaction mixture was measured at 500 nm by an ultraviolet spectrophotometer. Extraction and quantification of total flavonoids was implemented as previously described [44]. Flavonol content was estimated by high performance liquid chromatography as described previously by Jakobek et al., 2007 [45]. The absorbance was measured at 510 nm for flavonoids and at 425 nm for flavonols, respectively. Six biological replicates and at least three technical replicates were used for each measurement. Fruit color was measured with a Hunter Lab Mini Scan XE Plus colorimeter at four equatorial positions evenly distributed around the fruit.

### 2.3. Quantitative Real-Time RT-PCR (qRT-PCR) Analysis

Total RNA was extracted and purified from sweet cherry fruits using an EASYspin RNA Plant RNA rapid extraction kit (Yuanpinghao Bio, Tianjin, China) according to the manufacturer’s protocol. The samples triturated with liquid nitrogen were dissolved in the lysis buffer, and the supernatant was collected by centrifugation. After the supernatant passed through the adsorption column, the adsorption column was eluted with RNase-free water. First-strand cDNA was synthesized from purified RNA samples using a FastQuant RT (With gDNase) kit (Tiangen Bio, Beijing, China) following the manufacturer’s instructions. qRT-PCR amplifications were carried out using cDNA templates with TransStart Top Green qPCR SuperMix (TransGen Biotech, Beijing, Chain) containing SYBR green fluorescent intercalating dye on an ABI7500 PCR thermocycler (Applied Biosystems, Foster City, CA, USA). The sweet cherry *actin* gene (Pav_sc0002247.1_g030.1.mk) was used as an internal control. Relative gene expression was calculated using the 2^−∆∆Ct^ method. The gene-specific primers used for qRT-PCR analysis are detailed in Appendix A. Six biological replicates and three technical replicates were used for the qRT-PCR analyses.

### 2.4. Plasmid Construction and VIGS in Sweet Cherry Fruits

A 426-bp *PavGST1* fragment was amplified and cloned into a pTRV2 vector, and the generated recombinant plasmid was transformed into *Agrobacterium tumefaciens* strain GV3101. VIGS in sweet cherry fruit was carried out as previously described [46,47]. *Agrobacterium tumefaciens* GV3101 strains expressing pTRV1 and pTRV2 or pTRV2-*PavGST1* vectors were mixed in a 1:1 ratio and infiltrated using needleless syringes into the basal pedicel of cherry fruits 25 days after full bloom until the whole fruit was permeated. At 15 dpi, 10 random sweet cherry fruits per strain were collected and photographed. The TRV-mediated gene silencing assays, which involved 50 transformed fruits per strain from the same tree each time, were performed with six biological replicates using six 10-year-old trees.

### 2.5. Vector Construction and Transient Overexpression in Sweet Cherry Fruits

Transient overexpression assays were performed in sweet cherry fruits using the *Agrobacterium tumefaciens* method. Briefly, full-length *PavGST1* coding sequences were amplified from sweet cherry fruit cDNA and cloned into a pCAMBIA3301 vector. The recombinant plasmids (pCAMBIA3301-PavGST1) and the negative control vector (pCAMBIA3301) were introduced into *Agrobacterium tumefaciens* GV3101 and then infiltrated into sweet cherry fruits. *Agrobacterium* strain GV3101 was cultured overnight at 28 °C to an OD_600_ of 0.8–1.0, resuspended in *Agrobacterium* infiltration buffer (10 mM MgCl_2_, 10 mM MES (pH 5.6), and 100 µM acetosyringone) to a final OD_600_ of 0.8, and then infiltrated into sweet cherry fruits at 35 days after full bloom. The fruits were collected and evaluated at 2 and 3 days after infiltration. Transient overexpression assays were carried out with three independent biological replicates, with at least 20 sweet cherry fruits used per replicate.

### 2.6. Yeast One-Hybrid (Y1H) Assay

A Y1H assay was performed using the Matchmaker Gold Yeast One-Hybrid system (Clontech, Palo Alto, CA, USA) according to the manufacturer’s instructions. The promoter of PavGST1 was cloned into a pAbAi vector to generate a pAbAi-PavGST1 recombinant plasmid, which was then linearized and inserted into the Y1H Gold yeast strain. The coding sequences of PavMYB10.1 and PavMYB75 were amplified and inserted into a pGADT7 vector to respectively generate pGADT7-PavMYB10.1 and pGADT7-PavMYB75 recombinant plasmids. The pGADT7-PavMYB10.1 and pGADT7-PavMYB75 recombinant plasmids were separately transformed into strain Y1HGold containing pAbAi-PavGST1. The resulting yeasts were incubated on SD/-Leu medium supplemented with aureobasidin A (AbA). Each DNA–protein interaction assay consisted of three biological replicates.

### 2.7. Dual-Luciferase Reporter Assay

Dual-luciferase reporter assays were performed in *N. benthamiana* leaves as previously described [48]. To form the reporter construct, the promotors of the *PavGST1* gene, pPavGST1, pPavGST1m1, pPavGST1m2, and pPavGST1m3, were amplified and independently inserted into pGreenII 0800-LUC vectors. The full-length ORFs of PavMYB10.1 and PavMYB75 were individually cloned into pGreenII 62-SK vectors to generate effector plasmids. *Agrobacterium tumefaciens* GV3101 containing the constructed effector and reporter plasmids were co-infiltrated into *N. benthamiana* leaves. The effector plasmid with an empty vector pGreenII 62-SK was used as a control. Two days later, a Dual-Luciferase Reporter Assay kit (Promega, Madison, WI, USA) was used to measure the activity of LUC and REN with a Luminoskan Ascent microplate luminometer (Thermo Fisher Scientific, Waltham, MA, USA) according to the manufacturer’s instructions. The dual-luciferase reporter assay was carried out with at least four biological replicates.

### 2.8. GUS Activity Analysis

The full-length coding sequences of PavMYB10.1 and PavMYB75 were separately cloned into *pCAMBIA1302* to generate the effector plasmids. The promoter fragment of *PavGST1* was inserted into pBI121 in place of the 35S promoter coding region to generate the reporter plasmid. The recombinant effector plasmid and reporter plasmid were then used to co-transform tobacco leaves via *Agrobacterium tumefaciens* strain GV3101. Two days after infiltration, GUS activity was measured as previously described [49]. The assay included at least three biological replicates.

### 2.9. Statistical Analysis

All data are presented as means ± standard deviation of at least three independent experiments. Statistically significant differences between means were determined using a *t*-test or one-way analysis of variance (ANOVA) at the 5% significance level in SPSS version 17.0 (SPSS Inc., Chicago, IL, USA). Graphs were constructed with Origin 9.1 (Microcal Software Inc., Northampton, MA, USA).

## 3. Results

### 3.1. Anthocyanin Accumulation in Sweet Cherry Fruit Skin during Fruit Development and Ripening

To investigate the basis of fruit skin coloration in sweet cherry, we evaluated the phenotypes and flavonoid composition of three cultivars with different fruit skin colors: dark red Chunlu, blush Hongshouqiu, and yellow 13–33. The developmental process of Chunlu, Hongshouqiu, and 13–33 sweet cherry fruits is shown in Figure 1A. During early fruit growth and development (1–4 weeks after full bloom (WAFB)), fruit skin color did not significantly differ among Chunlu, Hongshouqiu, and 13–33. After the yellow-white stage of sweet cherry fruit growth and development (5 WAFB), significant differences in fruit skin color were observed among the three cultivars (Figure 1A). The anthocyanin content of Chunlu increased rapidly with fruit development (from 5 to 9 weeks) and was higher than in Hongshouqiu and 13–33. The anthocyanin content of Hongshouqiu was low and increased slowly from 5 to 9 weeks, whereas no anthocyanins were detected in 13–33 (Figure 1B). Similarly, the proanthocyanidin and especially total flavonoid contents of Chunlu fruits were significantly higher than those of Hongshouqiu and 13–33 during later fruit development and ripening (7–9 WAFB). No significant difference in flavonol contents was detected among Chunlu, Hongshouqiu, and 13–33 (Figure 1C). These results suggest that differences in fruit skin color in sweet cherry are due to variations in anthocyanin accumulation.

### 3.2. Strong Correlation between PavGST1 Transcript Levels and Anthocyanin Accumulation in Sweet Cherry

To investigate the correlation between GSTs and anthocyanin accumulation, we examined transcript levels of 53 sweet cherry *GST* genes at different stages in the cultivars Chunlu, Hongshouqiu, and 13–33 by quantitative real-time RT-PCR (qRT-PCR). Among the 53 PavGSTs, Pav_sc0001124.1_g450.1.mk (named PavGST1) was more highly expressed in Chunlu fruits than those of Hongshouqiu and 13–33, and fruit transcript levels of *PavGST1* in Hongshouqiu were significantly higher than in 13–33 (Figure 2). Moreover, the expression level of PavGST1 rapidly and gradually increased during later fruit development and ripening (5–9 WAFB) in Chunlu and Hongshouqiu fruits, respectively, consistent with the anthocyanin contents of their fruit skins. In contrast, *PavGST1* expression was not detected during fruit growth, development, or ripening in sweet cherry cultivar 13–33 (Figure 2). These results indicate that PavGST1 probably participates in anthocyanin accumulation in sweet cherry.

### 3.3. Significantly Decreased Anthocyanin Accumulation in Sweet Cherry Fruits following Silencing of PavGST1 by VIGS

To determine whether PavGST1 is essential for anthocyanin accumulation in sweet cherry fruits during fruit growth and development, we performed virus-induced gene silencing (VIGS) to knock down the expression of the *PavGST1* gene in sweet cherry fruits. According to qRT-PCR, the expression of *PavGST1* was dramatically reduced in PavGST1-silenced fruits compared with control fruits (TRV) at 15 days post-inoculation (dpi), thus indicating that *PavGST1* was efficiently silenced (Figure 3A). Compared with control fruits, *PavGST1-*silenced fruits had visible fruit skin color defects at 15 and 25 dpi (Figure 3B). The epidermis of *PavGST1-*silenced fruits was green-yellow or pink around the injection site at 15 and 25 dpi, whereas that of control fruits was dark or blackish red (Figure 3B). This result indicates that transient knockdown of *PavGST1* dramatically reduced sweet cherry fruit coloration. At the same time, anthocyanin contents around the injection site of *PavGST1*-silenced fruits were significantly decreased compared with those of control fruits at 15 and 25 dpi (Figure 3C). These results suggest that PavGST1 plays an important role in anthocyanin accumulation in sweet cherry.

### 3.4. Downregulation of Anthocyanin Biosynthetic Structural Genes in PavGST1-Silenced Sweet Cherry Fruits

To further understand the role of PavGST1 in anthocyanin accumulation in sweet cherry, we used qRT-PCR to measure transcript levels of key structural genes in the anthocyanin biosynthetic pathway, including *PavPAL*, *PavC4H*, *Pav4CL*, *PavCHS*, *PavCHI*, *PavF3H*, *PavFLS*, *PavDFR*, *PavANS*, *PavLAR*, *PavANR*, and *PavUFGT*, in *PavGST1*-silenced fruits and control fruits at 15 and 25 dpi. As shown in Figure 4, the expressions of early structural genes in the anthocyanin biosynthesis pathway, including *PavPAL*, *PavC4H*, *Pav4CL*, *PavCHS*, *PavCHI*, and *PavF3H*, were significantly downregulated in *PavGST1*-silenced fruits compared with control fruits at 15 and 25 dpi (Figure 4). In contrast, transcription of the flavonol-biosynthesis key structural gene *PavFLS* was upregulated by the silencing of *PavGST1*, particularly at 25 dpi (Figure 4). Compared with the controls, transcript levels of late biosynthetic genes that specifically contribute to anthocyanin biosynthesis, namely, *PavDFR*, *PavANS*, and *PavUFGT*, were significantly downregulated in *PavGST1*-silenced sweet cherry fruits. In contrast, expression levels of the late PA biosynthetic genes *PavLAR* and *PavANR* were slightly higher in *PavGST1*-silenced sweet cherry fruits than in control fruits at 15 and 25 dpi (Figure 4).

### 3.5. Promotion of Anthocyanin Biosynthesis in Sweet Cherry Fruits by Transient Overexpression of PavGST1

To further validate that *PavGST1* is essential for sweet cherry fruit coloration, we transiently overexpressed this gene in sweet cherry fruits. The overexpression recombinant plasmid p35S::*PavGST1* and the pCAMBIA3301 empty vector (as a control) were separately introduced into sweet cherry fruits via *Agrobacterium*-mediated transformation. Two days after inoculation, a clear red color was observed around the injection site of skins of *PavGST1* transiently overexpressed fruits, whereas no obvious red color was detected around the injection sites of control fruits (Figure 5A). Compared with the empty vector control, intense red pigmentation was observed around the injection site of skins of *PavGST1*-overexpressing sweet cherry fruits 3 days after infiltration (Figure 5A). The expression of *PavGST1* was markedly upregulated in sweet cherry fruits transiently overexpressing *PavGST1* compared with control fruits (Figure 5B). Additionally, anthocyanin contents as well as the abundance of *PavGST1* protein were significantly higher in sweet cherry fruits transiently overexpressing *PavGST1* compared with empty vector control fruits at 48 and 72 h after inoculation (Figure 5C). We also measured transcript levels of anthocyanin biosynthetic structural genes by qRT-PCR. Transcript levels of the structural genes *PavPAL*, *PavC4H*, *Pav4CL*, *PavCHS*, *PavCHI*, *PavF3H*, *PavDFR*, *PavANS*, and *PavUFGT* were dramatically higher in *PavGST1*-overexpressing fruits than in control fruits, whereas expression levels of proanthocyanidin or flavonol biosynthetic structural genes *PavLAR*, *PavANR*, and *PavFLS* were almost the same between fruits transiently overexpressing *PavGST1* and control fruits (Figure 5D). These results further confirm that *PavGST1* promotes anthocyanin accumulation in sweet cherry.

### 3.6. Regulation of PavMYB10.1 and PavMYB75 Expression Levels by PavGST1 and Their Direct Binding to the PavGST1 Promoter

Multiple studies have demonstrated that four R2R3-MYB TF family members, namely, AtMYB75/PAP1, AtMYB90/PAP2, AtMYB113, and AtMYB114, regulate anthocyanin biosynthesis in *Arabidopsis*. The MYB-bHLH-WD40 (MBW) protein complex has been confirmed as central to the regulation of anthocyanin accumulation in many plant species. To evaluate the interaction between PavGST1 and PavMYBs, we analyzed expression levels of four *Arabidopsis* homologous anthocyanin-related R2R3-MYB TF family members—*PavMYB75/PAP1* (Pav_sc0000464.1_g100), *PavMYB10.1* (Pav_sc0000464.1_g130), *PavMYB113* (Pav_sc0000464.1_g250), and *PavMYB114* (Pav_sc0000464.1_g210), as well as *PavbHLH3* (Pav_sc0000146.1_g300) and *PavWD40* (Pav_sc0000138.1_g560)—in *PavGST1*-silenced sweet cherry fruits and control fruits or in *PavGST1*-overexpressing fruits and control fruits. No *PavMYB113* or *PavMYB114* gene expression was detected in sweet cherry fruits (Appendix A). Interestingly, *PavMYB10.1* expression was more highly upregulated in *PavGST1*-overexpressing fruits than control fruits; in contrast, transcript levels of *PavMYB10.1* in *PavGST1*-silenced sweet cherry fruits were significantly lower than in control fruits (Figure 6A). Moreover, *PavMYB75* exhibited expression levels similar to those of *PavMYB10.1* in *PavGST1*-silenced sweet cherry fruits and *PavGST1*-overexpressing fruits, whereas the expressions of *PavbHLH3* and *PavWD40* did not significantly differ between *PavGST1*-silenced sweet cherry fruits or *PavGST1*-overexpressing fruits and control fruits (Figure 6A).

To further explore the correlation between PavMYB10.1 or PavMYB75 and PavGST1, the 2500 bp upstream region of the *PavGST1* promoter was cloned and then analyzed using PlantCARE online tools. Predicted and putative *cis*-regulatory elements of the PavGST1 promoter region, such as the TATA-box, CAAT-box, A-box, and G-box, are listed in Appendix A. We also detected several major light-responsive elements (G-Box, GATA-box, GT1-motif, and TCCC-motif); hormone-responsive elements, including methyl jasmonate (MeJA)-responsive elements (CGTCA-motif, TGACG-motif), ABA-responsive elements (ABRE, TGACG-motif), and an auxin-responsive element (TGA-element); and defense and stress-responsive elements (TC-rich repeats), which suggests that the expression of PavGST1 is controlled by light, hormones, and abiotic stresses. In addition, three MYB binding site (MBS) elements (CAACCA) were located at −1464 bp, −1781 bp, and −2154 bp upstream of the transcriptional start site (Appendix A).

Next, yeast one-hybrid assays were performed to assess whether PavGST1 is directly regulated by PavMYB10.1 and PavMYB75 in sweet cherry. No growth of transformant yeast cells co-expressing the PavGST1 promoter and the AD-Empty vector was observed on SD/-Leu medium supplemented with AbA. In contrast, the resistance reporter gene AbA driven by the PavGST1 promoter and the PavMYB10.1- or PavMYB75-AD recombinant plasmid were co-expressed in yeast cells, and the yeast transformant cells were able to grow on SD/-Leu medium supplemented with AbA (Figure 6B). These results indicate that PavMYB10.1 and PavMYB75 directly bind to the promoter regions of PavGST1 in vitro.

### 3.7. Direct Activation of PavGST1 Expression by Both PavMYB10.1 and PavMYB75

To validate whether both PavMYB10.1 and PavMYB75 affect the transcriptional activity of PavGST1, we performed GUS activity and dual-luciferase (Luc) transient expression assays. For the GUS assay, the reporter plasmid PavGST1-Pro::GUS and the effector plasmids p35S::PavMYB10.1, p35S::PavMYB75, or p35S::PavMYB10.1-PavbHLH3 were constructed and transformed into tobacco leaves (Figure 7A). GUS activities were significantly higher in tobacco leaves co-transformed with PavGST1-Pro-GUS/CaMV35S-PavMYB10.1, PavGST1-Pro-GUS/CaMV35S-PavMYB75, or PavGST1-Pro-GUS/CaMV35S-PavMYB10.1-PavbHLH3 than in leaves co-transformed with PaPG1_Pro_-GUS/CaMV35S-Empty; this was especially the case for leaves co-transformed with PavGST1-Pro-GUS/CaMV35S-PavMYB10.1-PavbHLH3 (Figure 7B). In addition, PavMYB10.1 co-transformed into tobacco leaves exhibited higher transactivation effects on the PavGST1 promoter (Figure 7B) than did co-transformed PavGST1-Pro-GUS/CaMV35S-PavMYB75. We subsequently carried out a dual-luciferase (Luc) transient expression assay in tobacco leaves with a dual-reporter construct system containing PavGST1 promoter-driven LUC and CaMV35S promoter-driven Renilla luciferase (REN) and used an effector plasmid expressing PavMYB10.1, PavMYB75, or PavMYB10.1+PavbHLH3 driven by the CaMV35S promoter to determine the activation effect of PavMYB10.1 and PavMYB75 on PavGST1 expression (Figure 7C). Compared with the control, the LUC/REN ratio was significantly increased when the PavGST1 promoter-LUC reporter was co-transfected into tobacco leaves along with PavMYB10.1, PavMYB75, or PavMYB10.1+PavbHLH3 (Figure 7D); this was especially true for PavMYB10.1+PavbHLH3, which resulted in the highest LUC/REN ratio. Taken together, these results confirm that PavMYB10.1 and PavMYB75 can positively activate the transcription of *PavGST1*.

To explore whether the differential effects of PavMYB10.1 and PavMYB75 on the transcriptional activity of PavGST1 are due to distinct motifs in the PavGST1 promoter, we divided the full-length PavGST1 promoter into three fragments (PavGST1m1–PavGST1m3) relative to the three MBSs (Figure 7E) and carried out dual-luciferase assays to measure the PavGST1 promoter activity of each fragment. As shown in Figure 7F, changing the MBS1 element in the PavGST1 promoter from CAACCA to CTTCCT (PavGST1m1) resulted in significantly reduced transactivation activity by PavMYB10.1, but not by PavMYB75, relative to the non-mutated PavGST1 promoter (PavGST1). Similarly, mutation of the MBS3 element from CAACCA to CTTCCT in the PavGST1 promoter (PavGST1m3) significantly reduced (by 63%) the transactivation activity of PavMYB75 compared with the original PavGST1 promoter (Figure 7F), whereas the transactivation activity of PavMYB10.1 did not significantly differ between MBS3-mutated and non-mutated PavGST1 promoters (Figure 7F). Furthermore, PavMYB10.1 and PavMYB75 exerted no obvious transactivation activities when the MBS2 element of the PavGST1 promoter was changed from CAACCA to CTTCCT (PavGST1m2) (Figure 7F). These results confirm that PavMYB10.1 and PavMYB75 proteins specifically bind to MBS1 and MBS3, respectively, in the PavGST1 promoter.

## 4. Discussion

Fruit color is an important agronomic trait that largely determines sweet cherry fruit quality, and a red fruit skin color, which is mainly due to anthocyanin accumulation, is an important target in sweet cherry breeding [39]. Accumulated anthocyanins not only contribute to sweet cherry fruit coloration, but also protect against certain diseases and are beneficial to human health. Elucidation of the biosynthetic processes involved in anthocyanin accumulation would therefore be a major contribution to the improvement of sweet cherry fruit quality.

### 4.1. The Essential Role of PavGST1 Encoding GST in Anthocyanin Transport and Accumulation in Sweet Cherry

As is well known, anthocyanins are biosynthesized in the cytoplasm and accumulated in vacuoles. Previous research has revealed that GST-mediated transport plays a pivotal role in anthocyanin accumulation in many plant species, including maize [29], petunia [30], perilla [49], Arabidopsis [14,50], grape [33], litchi [35], apple [27], kiwifruit [28], and peach [37]. In the present study, we identified 53 GST genes and analyzed their expressions in three sweet cherry cultivars with different skin colors, namely, Chunlu, Hongshouqiu, and 13–33. We found a GST gene (PavGST1, Pav_sc0001124.1_g450.1.mk) whose expression was consistent with anthocyanin accumulation in fruit skins of the different sweet cherry cultivars, thus indicating that PavGST1 may play an important role in anthocyanin transport in sweet cherry. When we transiently overexpressed *PavGST1* and knocked down *PavGST1* expression in sweet cherry fruits, anthocyanin accumulations were significantly changed. Our results demonstrate that *PavGST1* encoding GST is essential for anthocyanin transport and accumulation. The functional role of GSTs in anthocyanin transport and accumulation has been ascertained by studying their loss-of-function mutants, such as *bz2* (bronze-2) from maize [29], *an9* (anthocyanin 9) from petunia [30], *fl3* (flavonoid 3) from carnation [31], *tt19* (transparent testa 19) from Arabidopsis [14], and *rap* (reduced anthocyanin in petioles) from strawberry [36].

At the same time, several GSTs in fruit tree species have been thoroughly studied through genetic approaches. These investigations in peach, apple, kiwifruit, and litchi have demonstrated that a GST member can be simultaneously responsible for anthocyanin sequestration and accumulation [27,28,35,37]. Heterologous expression of *PpGST1* [37] (*Prunus persica*), *VviGST4* [34] (*Vitis vinifera*), *LcGST4* [35] (*Litchi chinensis*), *FaRAP* [36] (*Fragaria ananassa*), *MdGSTF6* [27] (*Malus domestica*), and *AcGST1* [28] (*Actinidia chinensis*) in an *Arabidopsis* tt19 mutant has been found to functionally complement the anthocyanin-less phenotype in vegetative tissues, thus indicating that these anthocyanin-related GST genes possess similar, conserved functions in the regulation of anthocyanin transport and accumulation. Our findings similarly indicate that transient overexpression of *PavGST1* promotes anthocyanin accumulation in sweet cherry fruits.

Our study has also revealed that the major flavonoids in sweet cherry fruits are proanthocyanidins, anthocyanins, and flavonols and that (especially) flavonol and anthocyanin contents differ significantly among sweet cherry cultivars with different fruit skin colors. Among the 53 analyzed *GST* genes, *PavGST1* was the only one whose expression was highly correlated with different sweet cherry fruit skin colors. We hypothesize that *PavGST1* is related to both anthocyanin and proanthocyanidin accumulation. In Arabidopsis, TT19 participates in the transport and accumulation of both anthocyanins and proanthocyanidins, which has been inferred from the loss of TT19 function in an Arabidopsis mutant phenotype [51]. VviGST4 and AcGST1 are involved in both anthocyanin and proanthocyanidin accumulation as well, as these two genes functionally complement anthocyanin-less and PA-deficient phenotypes in vegetative tissues and seed coats, respectively, of the Arabidopsis *tt19* mutant [28,34].

### 4.2. A Putative Model for PavGST1 Expression during Sweet Cherry Fruit Skin Coloration

Several previous studies have revealed that MYB1/A/10 (an MYB transcription factor) can directly bind to anthocyanin transport-related GST gene promoters and activate their transcription. In *Arabidopsis*, overexpression of the *PAP1* gene (a R2R3-MYB transcription factor) activates most genes related to anthocyanin biosynthesis and transportation, including *AtTT19*, *AtDFR*, and *AtUFGT* [52]. In apple, MdMYB1 directly regulates and activates *MdGSTF6* expression [27]. In peach, PpMYB10.1 specifically recognizes the MBS1 element of the *PpGST1* promoter to directly activate the expression of this gene to regulate anthocyanin accumulation [37]. In strawberry, the expression of FvRAP is dramatically induced by the overexpression of FvMYB10 and significantly reduced in the fruit of *myb10* varieties [36]. Similar results have also been observed in tea, lilies, litchi, and kiwifruit [35,36,53,54]. According to our study, both PavMYB10.1 and PavMYB75 can directly activate the promoter activity of PavGST1. In addition, we found that PavMYB10.1 and PavMYB75 proteins specifically bind to different MBS elements in the PavGST1 promoter sequence, which suggests that expression of PavGST1 is regulated via different R2R3 MYB transcription factors. Furthermore, transient overexpression of *PavGST1* in sweet cherry accelerates PavMYB10.1 and PavMYB75 transcription. We thus speculate that the coordinated roles of PavGST1 in anthocyanin transport and PavMYB10.1 and PavMYB75 in anthocyanin biosynthesis ultimately control sweet cherry fruit skin coloration [55,56]. In strawberry, overexpression of *FvMYB10* in the *myb10^−^rap^−^* double-mutant blocks red color development [36], which suggests that the function of *FvMYB10* in fruit anthocyanin accumulation depends on *FvGST/RAP*.

On the basis of the above-mentioned results, we propose a functional model to explain the role of PavGST1 in anthocyanin accumulation in sweet cherry fruits (Figure 8). In our model, two R2R3-MYB transcription factors, PavMYB10.1 and PavMYB75, directly regulate PavGST1 expression by specifically binding to different MBSs of the PavGST1 promoter region and act as positive regulators of PavGST1 to promote anthocyanin accumulation. PavGST1 can accelerate the expression of PavMYB10.1 and PavMYB75, which might enhance anthocyanin accumulation. We conclude that PavMYB10.1 not only controls anthocyanin synthesis but also regulates anthocyanin transport in sweet cherry. Additional research is needed to determine whether the *PavMYB75* gene regulates anthocyanin accumulation in sweet cherry.

In conclusion, the transient silencing and overexpression analyses of PavGST1 in this study have revealed that PavGST1, a GST gene, plays an indispensable role in anthocyanin transport and accumulation in sweet cherry fruits. More specifically, PavMYB10.1 and PavMYB75 directly bind to the promoter of PavGST1 to enhance its transcriptional activity. Our study has provided novel insights into the molecular mechanisms by which PavMYB10.1 and PavMYB75 regulate the activity of PavGST1 during anthocyanin transport and accumulation in sweet cherry.

## Figures and Tables

**Figure 1 cells-11-01170-f001:**
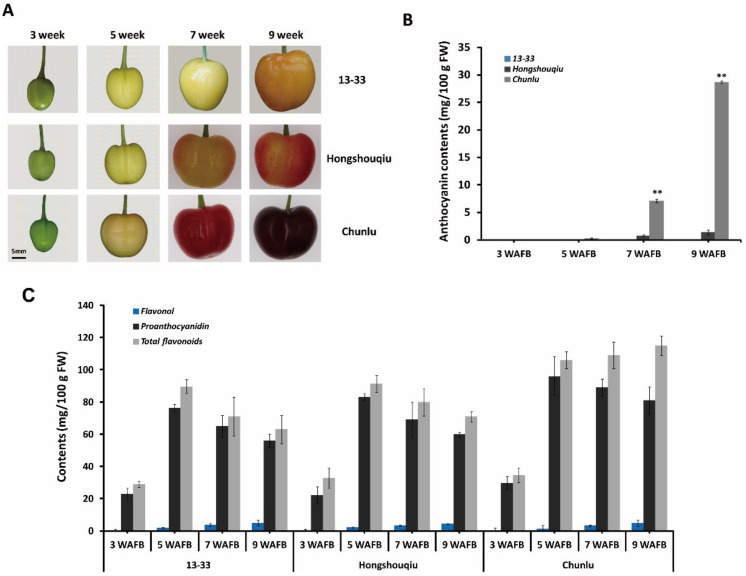
Developing fruits (**A**) and anthocyanin (**B**), proanthocyanidin (**C**), flavonol (**C**), and total flavonoid (**C**) contents measured 3, 5, 7, and 9 weeks after full bloom (WAFB) in sweet cherry varieties Chunlu, Hongshouqiu, and 13–33. Scale bar indicates 5 mm. Values are means ± SD from six independent biological replicates. ** Significant differences as calculated using the Student’s *t*-test at *p* < 0.01.

**Figure 2 cells-11-01170-f002:**
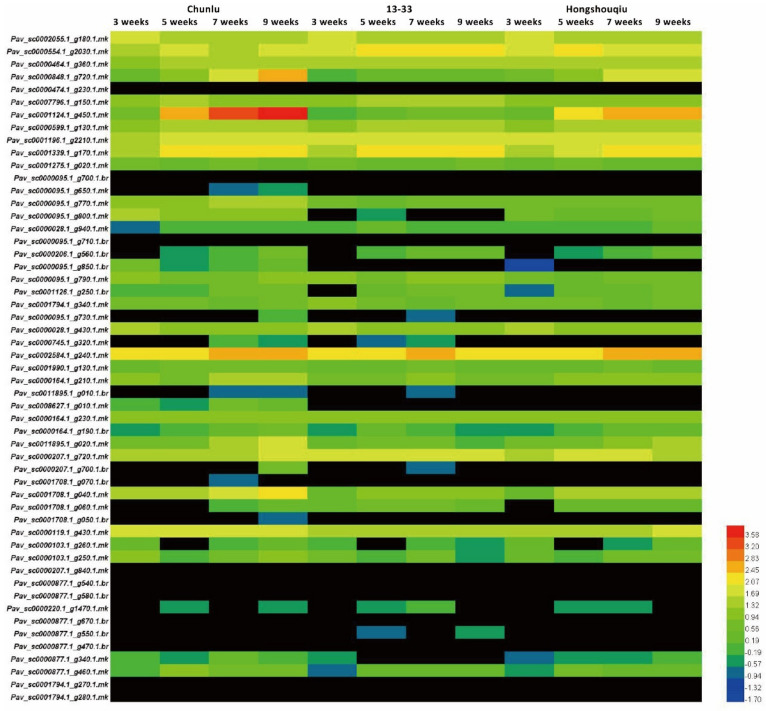
Expression patterns of 53 PavGST genes in developing fruits of sweet cherry varieties Chunlu, Hongshouqiu, and 13–33.

**Figure 3 cells-11-01170-f003:**
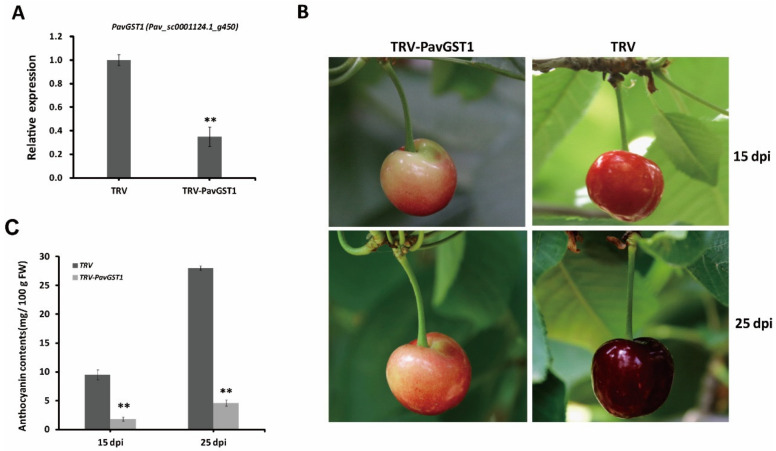
Effect of PavGST1 silencing on anthocyanin accumulation in sweet cherry fruits. (**A**) PavGST1 transcript levels in PavGST1*-*silenced and control fruits relative to *Pavactin* at 15 days post-inoculation (dpi), as determined by qRT-PCR. (**B**) Phenotypes of PavGST1*-*silenced and control fruits at 15 and 25 dpi in Chunlu sweet cherry varieties. (**C**) Changes in anthocyanin content in PavGST1*-*silenced and control fruits at 15 and 25 dpi. Values are means ± SD from six independent replicates. Statistical significance was determined according to one-way analysis of variance (ANOVA, **, *p* ˂ 0.01).

**Figure 4 cells-11-01170-f004:**
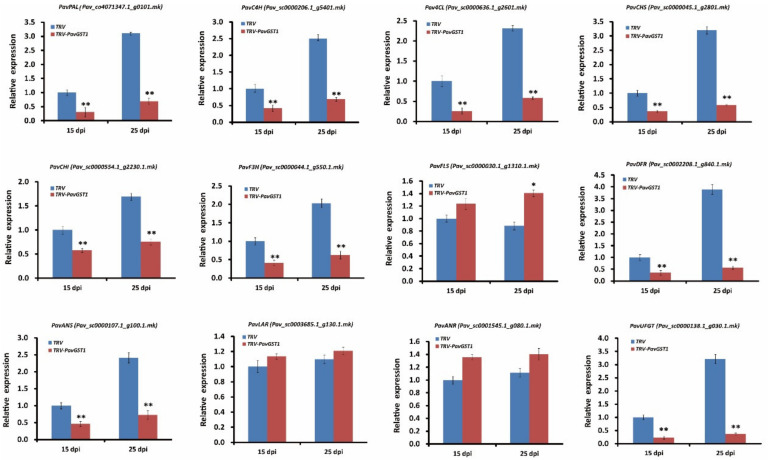
Transcript levels of anthocyanin pathway genes *PavPAL*, *PavC4H*, *Pav4CL*, *PavCHS*, *PavCHI*, *PavF3H*, *PavFLS*, *PavDFR*, *PavANS*, *PavLAR*, *PavANR*, and *PavUFGT* in PavGST1-silenced sweet cherry fruits and control fruits at 15 and 25 dpi. PAL, phenylalanine ammonia lyase; C4H, cinnamate 4-hydroxylase; 4CL, 4-coumarate-CoA ligase; CHS, chalcone synthase; CHI, chalcone isomerase; F3H, flavanone 3-hydroxylase; FLS, flavonol synthase; DFR, dihydroflavonol 4-reductase; ANS, anthocyanidin synthase, LAR, leucoanthocyanidin reductase; ANR, anthocyanidin reductase; UFGT, UDP-glucose:flavonoid-3-O-glucosyltransferase. Values are means ± SD from six independent replicates. Significant differences among means were determined by one-way analysis of variance (ANOVA, *, *p* < 0.05, **, *p* < 0.01).

**Figure 5 cells-11-01170-f005:**
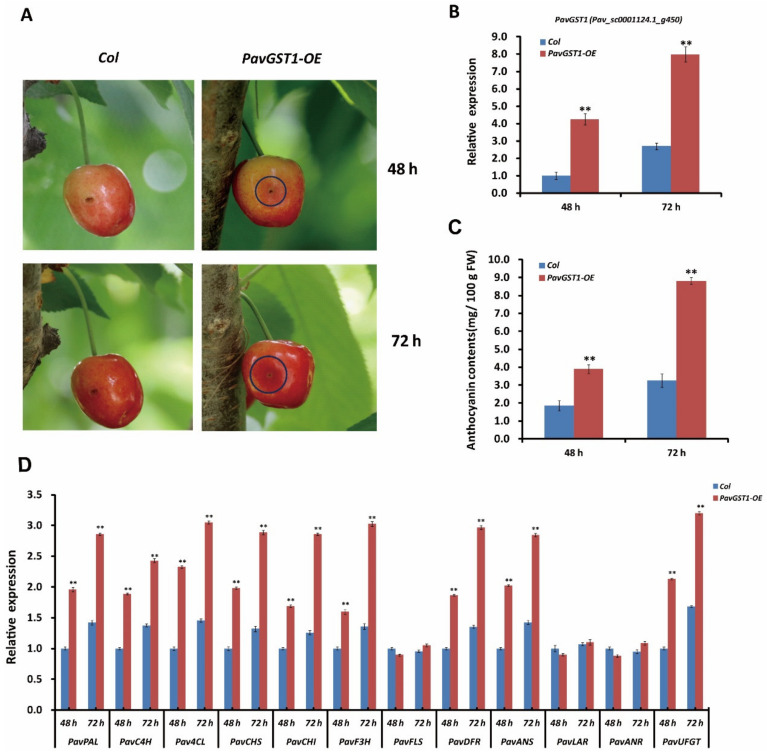
Enhanced anthocyanin accumulation in sweet cherry fruits resulting from transient expression of PavGST1. (**A**) Phenotypes of *PavGST1* transiently overexpressing fruits and control fruits recorded 48 and 72 h after inoculation in Chunlu sweet cherry varieties. (**B**) Results of qRT-PCR analysis of *PavGST1* expression levels in *PavGST1* transiently overexpressing fruits and control fruits. The *Pavactin* gene was used as an internal control. (**C**) Anthocyanin contents of *PavGST1* transiently overexpressing fruits compared with the control. (**D**) Transcript levels of anthocyanin pathway genes in *PavGST1* transiently overexpressing fruits and control fruits at 48 and 72 h after inoculation. Values represent means ± SD from six independent replicates. Statistical significance was determined using Student’s *t*-test (**, *p* ˂ 0.01).

**Figure 6 cells-11-01170-f006:**
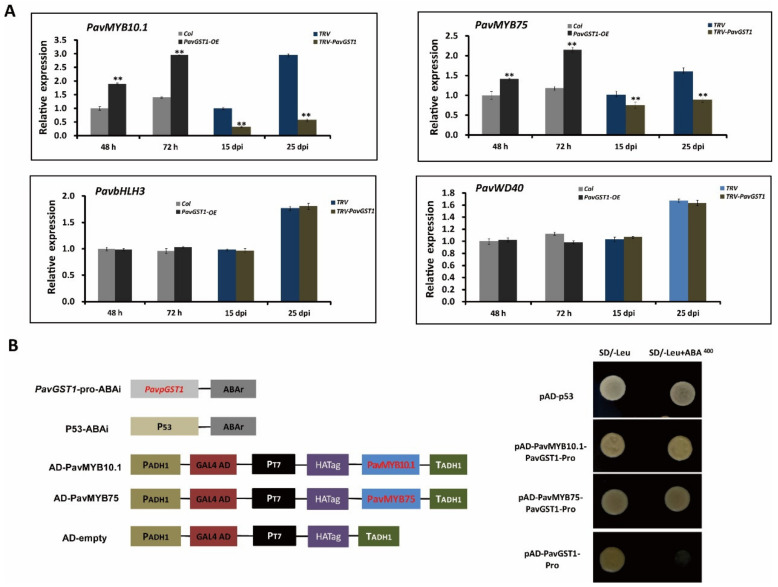
Data showing that PavMYB10.1 and PavMYB75 expression levels are regulated by PavGST1 and their direct binding to the PavGST1 promoter. (**A**) Changes in transcript levels of *PavMYB75*, *PavMYB10.1*, *PavbHLH3*, and *PavWD40* genes in *PavGST1*-overexpressing and PavGST1-silenced sweet cherry fruits. (**B**) Results of yeast one-hybrid (Y1H) assays showing PavMYB10.1 and PavMYB75 individually bound to the PavGST1 promoter. Values represent means ± SD from six independent replicates. Statistical significance was determined according to one-way analysis of variance (ANOVA, **, *p* < 0.01).

**Figure 7 cells-11-01170-f007:**
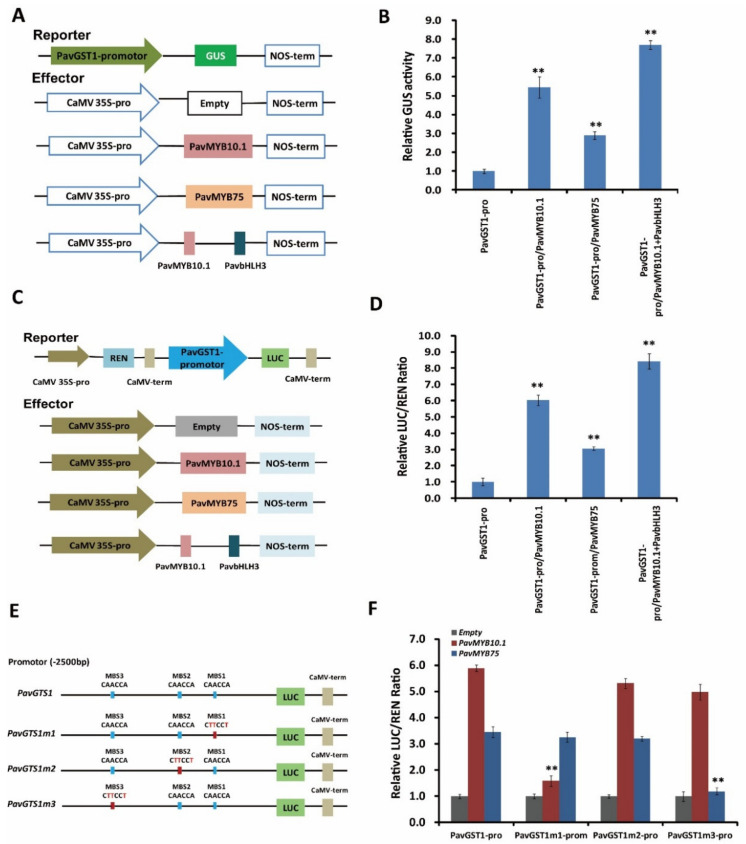
Direct activation of the PavGST1 promoter by PavMYB10.1 and PavMYB75. (**A**) Effector and reporter vector construction diagrams for GUS activity assays. (**B**) Results of GUS activity assays showing that PavMYB10.1, PavMYB75, and PavMYB10+PavbHLH3 all significantly activated the promoter of PavGST1. (**C**) Schematic representation of effector and reporter vectors used for dual-luciferase assays. (**D**) Effects of PavMYB10.1, PavMYB75, and PavMYB10+PavbHLH3 on PavGST1 promoter activity according to dual-luciferase assays. (**E**) Schematic diagram of PavGST1 promoter motif mutations. (**F**) Effects of PavMYB10.1 and PavMYB75 on the activities of original and mutated promoters of PavGST1 in dual-luciferase reporter assays. Each experiment included three replicates. Values represent means ± SD from six independent replicates. Significant differences were determined using Student’s *t*-test (**, *p* < 0.01).

**Figure 8 cells-11-01170-f008:**
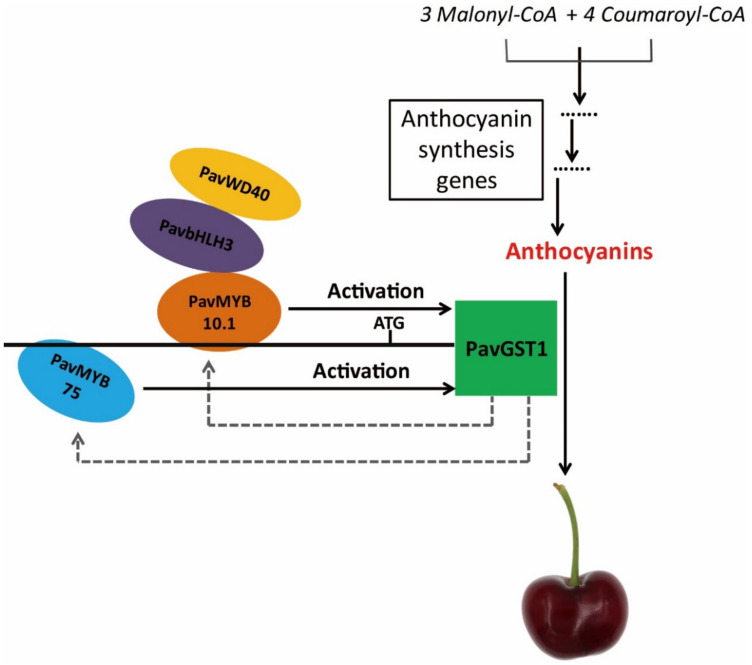
Proposed model of the role of PavMYB10.1 and PavMYB75 transcription factors in the positive regulation of anthocyanin accumulation by PavGST1 in sweet cherry. In this model, PavMYB10.1 and PavMYB75 bind to different MBS recognition sites of the PavGST1 promoter to activate PavGST1 transcription.

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
