# Peer review of "A Sweet Cherry Glutathione S-Transferase Gene, PavGST1, Plays a Central Role in Fruit Skin Coloration"

_cells, 2022, doi:10.3390/cells11071170_

Round 1

Reviewer 1 Report

Dear authors,

The comments and suggestions are attached.

Author Response

Dear Reviewer 1:

Thank you for arranging a timely review for our manuscript " A sweet cherry glutathione S-transferase gene, PavGST1, plays a central role in fruit skin coloration (cells-1622302)". We are pleased to know that our study is of general interest for the readers of Cells. We have carefully studied the reviewers’ critical comments and thoughtful suggestions, responded to these suggestions point-by-point, revised the manuscript accordingly, and upload the revised manuscript. We hope meet with their approval. We are greatly appreciated both your help and that of the reviewers concerning improvement to this paper. We hope the revised manuscript is suitable for publication.

With best regards

Sincerely yours Prof. Ming Li

Zhengzhou Fruit Research Institute,

Chinese Academy of Agricultural Sciences,

Zhengzhou, 450009, China

  • Question: The proposed theme “A sweet cherry glutathione S-transferase gene, PavGST1, plays a central role in fruit skin coloration” is current and relevant, but I missed innovation. The authors are encouraged to draw further attention to the innovation of the present study. What do the authors propose new with this study? What are the main features that have not yet been evaluated in previous studies?

Answer: Thank you for your helpful comments. GSTs play important roles in flavonoid accumulation. However, there are 53 GST gene families in the sweet cherry genome, which genes play a key role in the coloration of sweet cherry fruit has not yet been determined. Moreover, its expression regulation is also unknown. Therefore, it is necessary to carry out the functional research of sweet cherry GST family genes.

Our study has provided novel insights into the molecular mechanisms by which PavMYB10.1 and PavMYB75 regulate the activity of PavGST1 during anthocyanin transport and accumulation in sweet cherry, suggestting that there may be two different regulatory pathways for anthocyanin synthesis in sweet cherry.

Indeed, some points should be considered to improve the manuscript:

  • Question: Review the formatting of the Abstract, as well as the keywords, taking into account the rules of the Cells journal.

Answer: Thank you for your critical comments. We have revised the 'Abstract and keywords' section according to your suggestions. I greatly appreciate both your help and that of you concerning improvement to this paper. Special thanks to you for your good comments.

  • Question: Introduction: Some sentences are missing reference, e.g: Line 68: Sweet cherry (Prunus avium L. [Rosaceae]) is an economically important horticultural crop widely cultivated in temperate regions worldwide.

Answer: Thank you for your helpful comments. We have improved this part as your suggestion. Please see Line 87, Page 5.

Materials and Methods:

2.1. Plant materials

  • Question: Authors are encouraged to provide further information on sample collection. How about the fruit maturity parameters at harvest?

Answer: Thank you for your suggestion. We have improved this part. Please see Line 117, Page 6.

2.2. Anthocyanin, proanthocyanidin, flavonol, and total flavonoid extractions and measurements

  • Question: These determinations could be improved given more information. Both method of extraction, as well as the method of analysis.

Answer: Thank you for your helpful comments. We have revised the parts. Please see Line 124-137, Page 6-7.

2.3. Quantitative real-time RT-PCR (qRT-PCR) analysis

  • Question: “Total RNA was extracted and purified from sweet cherry fruits using an EASYspin RNA Plant RNA rapid extraction kit (Yuanpinghao Bio, Tianjin, China) according to the manufacturer’s protocol.” It is important give more information about the protocol, mainly the original reference. I suggest that you provide information for readers to refer to in the reference list.

Answer: Thank you for your suggestion. We have revised this part in our manuscript. Please see Line 142-145, Page 7.

  • Question: How about the fruit color determination?

Answer: Thank you for your suggestion. We have added this part. Please see Line 138-139, Page 7.

Results

  1. Results

3.1. Anthocyanin accumulation in sweet cherry fruit skin during fruit development and ripening

  • Question: The results should be expressed in table, with mean, standard deviation, p-value, and t-test 5%. In the Material and Methods section, the authors refer to “Six biological replicates and at least three technical replicates were used for each measurement”, on the other hand, in the Results section, it was mentioned: “Figure 1. Developing fruits (A) and anthocyanin (B), proanthocyanidin (C), flavonol (C), and total flavonoid (C) contents measured 3, 5, 7, and 9 weeks after full bloom (WAFB) in sweet cherry varieties ‘Chunlu’, ‘Hongshouqiu’, and ‘13-33’. Scale bar indicates 5 mm. Values are means ± SD from three independent biological replicates”

How many replicates were used in this study?

Answer: Six biological replicates were used in this study. Thank you for your critical comments. We have revised this part. Please see Line 899, Page 41.

  • Question: In the materials and Methods section, the authors mentioned used t-test, and along the manuscript, the authors said they used Student´s t-test. The authors should provide more information about the statistical analysis used in this manuscript. How about the one-way ANOVA test?

Answer: Thank you for your helpful comments. We have improved this part as your suggestion. Please see Line217, Page 11 and Figure captions.

  • Question: All figures must be improved. The authors proved a lot of information with low resolution and quality. The results expressed in this way are difficult to understand.

Answer: Thanks for your suggestion. We had improved Figures. Please see Figures.

  • Question: In my opinion, taking into account a research paper, 65 references are too much.

Answer: Thank you for your suggestion. We have changed the part. Please see "References".

Reviewer 2 Report

In the manuscript ‘A sweet cherry glutathione S-transferase gene, PavGST1, plays a central role in fruit skin coloration’, the authors revealed a PavGST gene, PavGST1, which played a central role in sweet cherry fruit anthocyanin accumulation, but there are some questions need to be revised.

In Figure 3B, the fruits used for TRV-mediated gene silencing assays were not uniform. The fruit used for TRV-PavGST1 was one of the cluster of fruits, while, fruit used for TRV was only one fruit. Besides, there was no needleless syringes trace in Fig 3B compared with Fig 5A. As we all known, the fruits do not ripen at the same time even in one same tree, so this result is unconvincing. 

While, GSTs act as one of anthocyanin transporters (line 42-44), why and how PavGST1 can influence the expressions of anthocyanin biosynthetic and regulatory genes? eg. Fig 4, Fig 5D.

As for the transcriptional regulation, yeast one-hybrid assay indicated PavMYB10.1 and PavMYB75 directly bind to the promoter regions of PavGST1 (Fig 6B), why the GUS activity and dual-luciferase (Luc) transient expression assays added PavbHLH3, and only added PavMYB10.1-PavbHLH3. Fig 7B, D

PavWD40 and PavbHLH3 in the model in Fig 8 was not suitable. Their results can’t support the conclusion.

Author Response

Dear Reviewer 2:

Thank you for arranging a timely review for our manuscript " A sweet cherry glutathione S-transferase gene, PavGST1, plays a central role in fruit skin coloration (cells-1622302)". We are pleased to know that our study is of general interest for the readers of Cells. We have carefully studied the reviewers’ critical comments and thoughtful suggestions, responded to these suggestions point-by-point, revised the manuscript accordingly, and upload the revised manuscript. We hope meet with their approval. We are greatly appreciated both your help and that of the reviewers concerning improvement to this paper. We hope the revised manuscript is suitable for publication.

With best regards

Sincerely yours Prof. Ming Li

Zhengzhou Fruit Research Institute,

Chinese Academy of Agricultural Sciences,

Zhengzhou, 450009, China

In the manuscript ‘A sweet cherry glutathione S-transferase gene, PavGST1, plays a central role in fruit skin coloration’, the authors revealed a PavGST gene, PavGST1, which played a central role in sweet cherry fruit anthocyanin accumulation, but there are some questions need to be revised.

  1. Question: In Figure 3B, the fruits used for TRV-mediated gene silencing assays were not uniform. The fruit used for TRV-PavGST1 was one of the cluster of fruits, while, fruit used for TRV was only one fruit. Besides, there was no needleless syringes trace in Fig 3B compared with Fig 5A. As we all known, the fruits do not ripen at the same time even in one same tree, so this result is unconvincing.

Answer: Thank you for your suggestion. We agree and have changed. Please see Figure3. The fruit growth and development process of sweet cherry is short, the fruit is completely mature at 55 days after full bloom (DAFB). Agrobacterium tumefaciens GV3101 strains expressing pTRV or pTRV-PavGST1 vectors were infiltrated into cherry fruits at 25 days after full bloom. At 15 and 25 days post-inoculation (dpi), 10 random sweet cherry fruits per strain were collected and photographed. 60 fruits for TRV-PavGST1 and TRV strains per replicate come from the same tree of ten years old. Transiently overexpressed fruit is on a branch where full bloom and fruit development are exactly the same.

  1. Question: While, GSTs act as one of anthocyanin transporters (line 42-44), why and how PavGST1 can influence the expressions of anthocyanin biosynthetic and regulatory genes? eg. Fig 4, Fig 5D.

Answer: Thank you for your helpful comments. We speculate that altering the expression of PavGST1 may feedback to change PavMYB10.1 gene expression, as well as related anthocyanin biosynthetic genes expression, suggesting that PavMYB10.1 regulates anthocyanin biosynthesis as well as transport. Further mechanism analysis requires more efforts to implement.

  1. Question: As for the transcriptional regulation, yeast one-hybrid assay indicated PavMYB10.1 and PavMYB75 directly bind to the promoter regions of PavGST1 (Fig 6B), why the GUS activity and dual-luciferase (Luc) transient expression assays added PavbHLH3, and only added PavMYB10.1-PavbHLH3. Fig 7B, D

Answer: Thank you for your helpful suggestion. Previous studies have demonstrated that the PavMYB10.1-PavbHLH-PavWD40 (MBW) protein complex regulates anthocyanin biosynthesis in sweet cherry and that PavMYB10.1 plays a key role in determining fruit skin coloration. PavbHLH3 is phosphorylated to enhance PavMYB10.1 transcriptional activation activity, ultimately leading to anthocyanin accumulation. PavWD40 does not bind directly to the promoters of anthocyanin biosynthesis genes but regulates anthocyanin biosynthesis through the formation of the MBW complex with the MYB and bHLH transcription factors. therefore, we only added PavMYB10.1-PavbHLH3 in the GUS activity and dual-luciferase (Luc) transient expression assays.

  1. Question: PavWD40 and PavbHLH3 in the model in Fig 8 was not suitable. Their results can’t support the conclusion.

Answer: Thank you for your critical comments. We have revised Figure 8 according to your suggestions. Please see Figure 8.

Round 2

Reviewer 1 Report

Dear authors,

I could see an extensive revision of the manuscript and a significant improvement in the way the data was presented.

However, I still see some points that can be adjusted before accepting the document in this journal.

Explain why in some cases the authors used 3 replicas and 6 replicas:

Figure 1: Values are means ± SD from six independent biological replicates.

Figure 3: Values are means ± SD from three independent replicates

Line 129: I believe that starting the sentence with 0.5g is not the most appropriate way.

Line 131: Describe the HCl compound

Line 550: DISCUSSION - Discussion 

Line 599: as well as (English needs to be improved)

Line 601: references must be in order [28, 34]. Please, review all references.

Figures: express the number of samples as n = ...

The figure captions are too long and repetitive.

The references should be verify one by one.

Author Response

Dear Reviewer 1:

Thank you for arranging a timely review for our manuscript "A sweet cherry glutathione S-transferase gene, PavGST1, plays a central role in fruit skin coloration (cells-1622302)". We have carefully studied your critical comments and thoughtful suggestions, responded to these suggestions point-by-point, revised the manuscript accordingly, and upload the revised manuscript. All changes made to the text are in blue so that your may be easily identified. We are greatly appreciated your concerning improvement to this paper. We hope the revised manuscript is suitable for publication.

With best regards

Sincerely yours Prof. Ming Li

Zhengzhou Fruit Research Institute,

Chinese Academy of Agricultural Sciences,

Zhengzhou, 450009, China

Reviewer 1:

I could see an extensive revision of the manuscript and a significant improvement in the way the data was presented.

However, I still see some points that can be adjusted before accepting the document in this journal.

  1. Question: Explain why in some cases the authors used 3 replicas and 6 replicas:

Figure 1: Values are means ± SD from six independent biological replicates.

Figure 3: Values are means ± SD from three independent replicates

Answer: Thank you for your helpful comments. Sorry for the mistaken expression. Values are means ± SD from six independent biological replicates in Figure 1 and Figure 3. We had improved this part. Please see Figure 3 caption.

  1. Question: Line 129: I believe that starting the sentence with 0.5g is not the most appropriate way.

Answer: Thank you for your helpful comments. We have revised this part. Please see Line129, Page 7.

  1. Question: Line 131: Describe the HCl compound

Answer: We have changed. Please see Line 131, Page 7.

  1. Question: Line 550: DISCUSSION - Discussion

Answer: Changed. Thanks!

  1. Question: Line 599: as well as (English needs to be improved)

Answer: Thank you for your helpful comments. All authors and a native English speaker PhD. Lesley Benyon had critically read and edited our manuscript.

  1. Question: Line 601: references must be in order [28, 34]. Please, review all references.

Answer: Thank you for your suggestion. We have changed and checked the part in our manuscript. Please see line 581, page 27; line 615, page 30. Thanks!

  1. Question: Figures: express the number of samples as n = ...

Answer: Thank you for your helpful comments. Express the number of samples as n≥30.

  1. Question: The figure captions are too long and repetitive.

Answer: Thanks for your suggestion. We had improved this part. Please see Figure captions.

  1. Question: The references should be verify one by one.

Answer: Thank you for your suggestion. We have carefully checked the part. Please see "References".

Reviewer 2 Report

The manuscript has been improved a lot, while there still some questions need to be revised. 1. They used three sweet cherry cultivars ‘Chunlu’, ‘Hongshouqiu’, and ‘13-33’ in this work, which cultivar was used in Fig 3 and Fig 5? 2. Fig 2 is not clear enough, and it’s better to highlight PavGST1 in the whole 53 PavGST genes. 3. Some text errors need to be revised, eg. GST gene was not italic in many places ‘lines 80-83’, ‘lines 105-109’, ‘line 260’ etc.

Author Response

Dear Reviewer 2:

Thank you for arranging a timely review for our manuscript "A sweet cherry glutathione S-transferase gene, PavGST1, plays a central role in fruit skin coloration (cells-1622302)". We have carefully studied your critical comments and thoughtful suggestions, responded to these suggestions point-by-point, revised the manuscript accordingly, and upload the revised manuscript. All changes made to the text are in blue so that your may be easily identified. We are greatly appreciated your concerning improvement to this paper. We hope the revised manuscript is suitable for publication.

With best regards

Sincerely yours Prof. Ming Li

Zhengzhou Fruit Research Institute,

Chinese Academy of Agricultural Sciences,

Zhengzhou, 450009, China

Reviewer 2:

The manuscript has been improved a lot, while there still some questions need to be revised.

  1. Question: They used three sweet cherry cultivars ‘Chunlu’, ‘Hongshouqiu’, and ‘13-33’ in this work, which cultivar was used in Fig 3 and Fig 5?

Answer: We have improved this part as your suggestion. Thank you for your helpful comments. Please see Figure 3 and 5 captions.

  1. Question: Fig 2 is not clear enough, and it’s better to highlight PavGST1 in the whole 53 PavGST genes.

Answer: Thank you for your suggestion. We have highlighted the PavGST1 in Figure 2. Please see Figure 2.

  1. Question: Some text errors need to be revised, eg. GST gene was not italic in many places ‘lines 80-83’, ‘lines 105-109’, ‘line 260’ etc.

Answer: Thank you for your suggestion. We have changed and checked the part in our manuscript. Please see line 77-83, page 4; line 105-106, page 6; line 260, 264, 268, page 13; line 376-377, page 18; line 420-423, 427, 429-431,page 20; line 432-433, page 21. line 583-585, page 27-28.Thanks!